# Exploring Metabolomic Patterns in Type 2 Diabetes Mellitus and Response to Glucose-Lowering Medications—Review

**DOI:** 10.3390/genes14071464

**Published:** 2023-07-18

**Authors:** Mina Shahisavandi, Kan Wang, Mohsen Ghanbari, Fariba Ahmadizar

**Affiliations:** 1Department of Epidemiology, Erasmus MC University Medical Center Rotterdam, 3015 GD Rotterdam, The Netherlandsm.ghanbari@erasmusmc.nl (M.G.); 2Department of Data Science & Biostatistics, Julius Global Health, University Medical Center Utrecht, 3508 GA Utrecht, The Netherlands

**Keywords:** type 2 diabetes mellitus, metabolomics, metabolites, medications, glucose-lowering medications

## Abstract

The spectrum of information related to precision medicine in diabetes generally includes clinical data, genetics, and omics-based biomarkers that can guide personalized decisions on diabetes care. Given the remarkable progress in patient risk characterization, there is particular interest in using molecular biomarkers to guide diabetes management. Metabolomics is an emerging molecular approach that helps better understand the etiology and promises the identification of novel biomarkers for complex diseases. Both targeted or untargeted metabolites extracted from cells, biofluids, or tissues can be investigated by established high-throughput platforms, like nuclear magnetic resonance (NMR) and mass spectrometry (MS) techniques. Metabolomics is proposed as a valuable tool in precision diabetes medicine to discover biomarkers for diagnosis, prognosis, and management of the progress of diabetes through personalized phenotyping and individualized drug-response monitoring. This review offers an overview of metabolomics knowledge as potential biomarkers in type 2 diabetes mellitus (T2D) diagnosis and the response to glucose-lowering medications.

## 1. Introduction

Currently, more than half a billion adults are affected by diabetes globally, and the number is foreseen to rise to nearly 800 million by 2045, with the majority being type 2 diabetes mellitus (T2D) [1]. T2D is a multifactorial metabolic disorder marked by dysregulated glucose homeostasis, insulin resistance, and impaired insulin secretion. Poor glucose control in people with T2D results in micro and macrovascular complications, including retinopathy, neuropathy, nephropathy, and cardiovascular disorders [2]. Risk factors that cause T2D development consist of unfavorable dietary patterns, lifestyle, and genetic influences, which by interacting with each other, make disease prevention and treatment rather complex [3].

Allowing for enhancing the comprehension of the genetic makeup of T2D, over 700 T2D risk loci have been detected to be associated with T2D [4]. However, the genetic factors identified through genome-wide association studies (GWAS) contribute to the risk of T2D to a limited extent [5]. Identifying biomarkers for screening and predicting T2D and its complications can aid in personalized healthcare management. Furthermore, it can provide insights into the underlying pathways involved in the progression of T2D.

Personalized medicine, also known as precision medicine, aims to tailor medical treatments to individual patients based on their unique characteristics. It considers the complex interactions between genetic and environmental factors that contribute to inter-individual variations in therapeutic outcomes and disease susceptibility. While pharmacogenomics has made significant progress in correlating drug responses with genetic polymorphisms, it does not account for the impact of environmental factors and the co-metabolism of host and gut microbiota. Metabolomics, on the other hand, offers an integrative approach to studying the variation of endogenous metabolites in response to modifications in biological systems [6].

Metabolomics has emerged as a powerful and efficient technique in recent years for screening, diagnosis, and prognosis of T2D. Metabolomics is the systematic identification and quantitation of small-molecule “metabolites“ that integrate information from the genome, transcriptome, proteome, and enzymes in interaction with external factors (e.g., lifestyle) [7]. Techniques related to a metabolomics assessment usually include nuclear magnetic resonance (NMR), liquid chromatography-mass spectrometry (LC-MS), and gas chromatography-mass spectrometry (GC-MS) [8]. These techniques are used to identify and quantify metabolites from a biological sample. The recent rapid development of a range of analytical platforms, including GC, high performance liquid chromatography (HPLC), ultra-performance liquid chromatography (UPLC), and capillary electrophoresis (CE) coupled to MS and NMR spectroscopy, could enable separation, detection, characterization, and quantification of such metabolites and related metabolic pathways [9].

The metabolome comprises all the metabolites present within a biological sample, tissue, or organ, including signaling molecules, which serve as the final products of cellular processes. Metabolomics studies can be conducted through targeted or untargeted approaches, utilizing various techniques as explained earlier. Both targeted and untargeted metabolomics approaches have the potential to enhance our understanding of disease progression by identifying key pathways and discovering novel signaling molecules. Some of these molecules may have the potential to serve as biomarkers. With advancements in high-throughput technologies, metabolomics is increasingly being recognized as a valuable tool for biomarker discovery in the diabetes field [10].

Metabolomics provides a comprehensive readout of genetic and environmental influences. It has been successfully applied in various areas, such as biomarker discovery, mechanistic studies of diseases and drug activity, and evaluation of drug-induced toxicity and metabolism. By analyzing the relationship between these metabolites and treatment outcomes, researchers aim to identify biomarkers to guide medication selection. Another key objective is to optimize medication therapy based on individual patient characteristics and risk profiles, aligning with the principles of personalized medicine. This perspective emphasizes tailoring treatments to maximize effectiveness and minimize risks for each patient [6].

In this review, we aimed to briefly address the metabolomics signature of T2D (Table 1) and evaluate metabolomics related to optimizing treatment with glucose-lowering medications (Table 2).

## 2. Metabolomics Signature of Type 2 Diabetes Mellitus

In the last 20 years, metabolomics has been extensively utilized in epidemiological studies, leading to significant findings regarding metabolite pathways linked to developing T2D. This approach has allowed researchers to uncover and explore specific metabolic pathways that contribute to the underlying causes of T2D. By analyzing metabolite profiles, valuable insights have been gained into the metabolic alterations associated with T2D, shedding light on the disease’s etiology. These findings have the potential to enhance our understanding of T2D and provide new avenues for prevention, diagnosis, and targeted therapeutic interventions [7,16,32,33,34,35].

Plasma metabolites have been categorized into hydrophilic polar components (i.e., carbohydrates, nucleic acids, and amino acids) and hydrophobic non-polar components involving lipid metabolism (lipidomics) [36].In a recent systematic review and meta-analysis, the relationship between 412 metabolites present in blood or urine and the development of T2D was extensively examined. The study encompassed a comprehensive analysis of 61 individual studies involving 71,196 participants, among whom 11,771 developed incident T2D. By aggregating the data from these studies, the researchers aimed to gain a deeper understanding of the association between specific metabolites and the risk of developing T2D. This study indicates that several blood metabolites, including lipids and carbohydrates, are associated with T2D development [16]. Besides that, the increased levels of certain amino acids were associated with insulin resistance in non-obese individuals, suggesting that amino acid imbalances play a role in this condition independent of commonly accepted risk factors like circulating fatty acids and inflammatory cytokines [37].

Highlighting the significance of understanding the metabolic disturbances in T2D and its complications, recent advancements in metabolomics have enabled the identification of circulating biomarkers associated with T2D even before its onset. These biomarkers offer potential applications in the screening, diagnosis, and prognosis while providing insights into the underlying pathways involved in T2D development. Integrating the omics approaches with genomics can uncover causal associations, although careful utilization of these methods is necessary due to their limitations [33].

### 2.1. Lipids

Previous epidemiological studies have consistently demonstrated a long-standing association between lipids and diabetes. However, due to the lack of specificity for traditional clinical measurements, the lipid profile was the only available target for prior metabolomics studies for a long time. Thus, most prior studies investigating dyslipidemia among diabetes mainly focused on triglycerides and high and low-density lipoprotein cholesterol. These findings mainly supported the notion that low levels of high-density lipoprotein cholesterol (HDL-C) and high levels of triglyceride (TG) are significant predictors of the occurrence of T2D development in adults [14,15]. The low levels of HDL-C also greatly enhanced the tyrosine serum level in patients with T2D [38].

Based on an updated systematic review and meta-analysis of prospective cohorts encompassing a vast participant pool of 71,196 individuals, among whom 11,771 were diagnosed with T2D, demonstrated that 123 metabolites are significantly associated with T2D risk using high throughput metabolomics data. They concluded that several glycerolipids, (lyso)phosphatidylethanolamines, dihydroceramide, and ceramides are linked to an elevated risk of developing T2D [16].

In addition, a recent study among the Chinese population included 5731 people, of whom 529 participants developed T2D. This study investigated a panel of novel sphingolipids, including ceramides, saturated sphingomyelins, unsaturated sphingomyelins, hydroxyl-sphingomyelins, and hexosyl ceramide and their association with T2D incidence in a six-year follow-up study. The result indicated a positive association of these metabolites with incident T2D and β-cell dysfunction. According to this study, sphingolipids incorporate in developing impaired glucose homeostasis by inducing insulin resistance, impairing β-cell function, and inflammation [39].

Due to advances in high-throughput metabolomics technology and subtler lipid species or lipidomics analysis, we can now study the total acyl chain carbon number and degree of unsaturation of plasma lipids. Therefore, odd-chain saturated fatty acids (OCFA)-containing lipids were found to exhibit a sex-specific association with the risk of developing T2D. At the same time, specific OCFA-containing Phospholipids, such as Phosphatidylcholine C15:0, were only negatively correlated with the risk of developing T2D in women but not men [18].

The results of a case-control study involving 107 men with T2D and 216 controls sourced from the longitudinal METSIM study indicated higher levels of triacylglycerols, di-acyl-phospholipids, lysoalkylphosphatidylcholine, and lysophosphatidylcholine acyl and that lower levels of alkyl-acyl phosphatidylcholines are linked to the higher risk of T2D [19]. The models of lipids remained reliable for the development of T2D within the fasting plasma glucose-matched subset even in the validation phase of the study. This study demonstrates that a characteristic lipid molecule of T2D exists many years before diagnosis and enhances the likelihood of progression to T2D. Besides that, other endeavours reported that diacyl-phosphatidylcholines (C32:1, C36:0, C36:1, C38:3, and C40:5) were significantly altered in T2D compared to non-T2D subjects [20,40]. By repeatedly measuring plasma lipid metabolites at the baseline of 250 incident T2D cases and 692 participants without T2D at the baseline after one year of follow-up, the PREDIMED trial found that the plasma lipid profiles composed of triacylglycerols, diacylglycerols, and phosphatidylethanolamines were associated with a higher risk of T2D [17]. These fatty acids (FAs) are mainly derived from dietary triglycerides and phospholipids. Thus, they may be targeted as new interventions in diabetes dietary prevention.

### 2.2. Amino Acids

Among various amino acid metabolites, branch-chain amino acids (BCAAs) (i.e., leucine, isoleucine, and valine) and their related metabolites have been reported to be strongly associated with early diagnosis and predicting the occurrence of T2D [11,16,41,42,43]. BCAAs have an impact on various cellular signaling pathways and their association with insulin resistance. BCAAs are known to enhance protein synthesis and increase mitochondrial content in muscle and adipocytes. However, elevated circulating BCAA levels have been linked to insulin resistance, potentially due to dysregulated BCAA degradation [44]. BCAAs—particularly leucine—activate the rapamycin complex1 mTORC1, which is known to regulate cell growth and metabolism, glucose metabolism, and several more essential physiological processes [45]. When incorporated into a dietary pattern that includes high-fat consumption, BCAAs contribute to the development of obesity, insulin resistance, and diabetes. While BCAAs have been linked to anti-obesity effects, higher circulating levels of BCAAs are observed in individuals with obesity. They are associated with poorer metabolic health and increased risk of insulin resistance and T2D. Insulin resistance may also contribute to elevated levels of amino acids by promoting protein degradation and impairing BCAA oxidative metabolism in certain tissues [46,47]. Furthermore, results from the Framingham Heart Study (FHS) found that participants with higher serum levels of BCAAs had a higher risk of developing T2D even after adjusting for the body mass index (BMI) [48]. This might be explained by altering cellular insulin signaling due to increased serum BCAAs levels and involving the mammalian rapamycin pathway, pancreatic islet β-cells, and adipocytes, leading to cytotoxic metabolite build-up [49,50].

Besides that, imbalances in amino acid homeostasis are also linked to insulin resistance among people with low BMI and higher levels of the aromatic amino acids (AAAs) tyrosine,2-hydroxybutyrate, methionine, phenylalanine, lysine, histidine, 2-aminoadipate, alanine, and glutamate have also been associated with an increased risk of T2D [12,37,42,51,52,53,54]. On the other hand, a negative association has been found in serum concentration of Glycine, Glutamine, and the risk of T2D development [11]. However, the results are less consistent for specific amino acids such as Glycine, which was inversely associated with incident T2D in Europeans [40] but with a positive association in the Chinese population [13]. Moreover, results from the Mendelian Randomization (MR) analysis embedded in the FHS Offspring cohort reported a negative association between glycine and T2D risk [12]. This lack of consistency may contribute to mainly including white participants in mentioned studies, and evidence from other ethnicities is needed to achieve generalizability [43,55,56].

In a similar study, the metabolite signatures of obese children with T2D, obese children without diabetes (OB), and healthy normal weight controls (NW) were compared. Using targeted LC-MS/MS, 22 urine metabolites were identified that were uniquely associated with T2D. The results included metabolites related to the betaine pathway, nucleic acid metabolism, and BCAAs. Moreover, urine levels of succinylaminoimidazole carboxamide riboside (SAICA-riboside) were found to be increased in diabetic youth, suggesting its potential as a biomarker for T2D [57].

### 2.3. Carbohydrates

According to an updated meta-analysis of the case-cohort studies (71,196 participants and 11,771 T2D cases/events), higher carbohydrate metabolites, including mannose and trehalose, are also associated with an increased risk of T2D. Furthermore, higher levels of glycine, glutamine, betaine, indolepropionate, and (lyso)phosphatidylcholines were associated with lower T2D risk (hazard ratio 0.69–0.90) [6].

This study showed a significant association between glycolysis/gluconeogenesis metabolite (i.e., pyruvate) and higher T2D risk [16]. Another systematic review involving 27 cross-sectional and 19 prospective cohort studies revealed that metabolites embedded in sugar, including glucose, hexose, mannose, arabinose, fructose, and glycolipids, were positively associated with the prevalence of T2D [58]. Additionally, using a non-targeted metabolomics platform in a case-control study (with 115 cases diagnosed with T2D, 192 individuals with impaired fasting glucose, and 1897 control subjects), results showed that subjects with lower plasma levels of 1,5-anhydroglucitol, and higher plasma concentrations of glucose, mannose, and fructose had more risk of impaired fasting glucose and T2D [22].

Complex interactions between genetic and environmental factors influence the metabolite profile of an individual. GWAS have been used to explore the impact of genetic variation on plasma metabolites. The identified genetic variants associated with metabolite levels, particularly enzymes and carriers involved in processes like β-oxidation, fatty acid and phospholipid biosynthesis, as well as amino acid metabolism. Notably, these genetic loci explain a significant portion of the variance in metabolites, highlighting the role of genetics. Furthermore, specific genetic variants have been linked to glycine, serine, and betaine levels, although their connection to diabetes-related traits remains unclear [59].

Environmental factors such as one’s diet, activity, medication, and the microbiome contribute to the complexity of the metabolome. Diet, physical activity, gender, and age influence metabolomic profiles and should be considered in study design and interpretation. The gut microbiome plays a significant role in host metabolism and metabolomics profiles, with obesity and T2D associated with altered microbial profiles and reduced diversity. Microbial populations can impact host metabolism, intestinal development, and insulin secretion and contribute to metabolic disorders. Microbiota transplantation studies in mice and limited human data suggest the potential for improving metabolic health through modulation of the microbiome [60,61,62].

## 3. Metabolomics Signature of Response to Glucose-Lowering Medications

Individual variations in genetic and environmental factors can cause differences in metabolic patterns, which could influence medication responses. Though a substantial proportion of chronic conditions related to T2D are associated with aging, poor glycaemic control still plays an essential role in the pathogenesis of macro and micro-vascular diseases [63]. Improving response to glucose-lowering medications in T2D, where healthcare expenditure for diabetes is among the highest, could yield significant promotion in quality of life and reduce health burden. Yet, favorable responses to such therapeutics are unstable, in which roughly half of T2D patients do not reach the desired glucose levels [64]. Several factors may contribute to individual differences in response to glucose-lowering medication, such as age at the onset of diabetes, gender, deterioration of β-cell function, and genetic variations [65]. Despite impressive achievements in pharmacogenomics studies over the past decades, other omics layers in poor glycaemic control remain understudied.

The utilization of the metabolomics approach has been employed to investigate the potential pathways affected by medications. In this context, metabolomics enables molecular understanding, aids potential therapeutic target discoveries, and enhances T2D management. Moreover, increased utilization of metabolomics in intervention trials may uncover the underlying mechanism of effective treatment of glucose-lowering medications. The first line of T2D treatment and most widely used medication is metformin, which improves glucose homeostasis and insulin sensitivity, while the underlying mechanisms are still not fully understood. To elaborate, one narrative study summarized that the metabolic profiles associated with various pathways were significantly influenced by metformin treatment, regardless of the metabolic condition [66].

A cross-sectional study (*n* = 698 individuals) utilized the targeted metabolomics data from the Nightingale platform of four separated Dutch cohorts. This study investigated the associations between amino acids, glycolysis measures, ketone bodies, fatty acids, and lipid concentrations with glycaemic control according to stratified HbA1c levels by different glucose-lowering medications (metformin, Sulfonylurea, and insulin). Their pooled results showed that 26 out of 162 metabolites were significantly associated with insufficient glycemic control (HbA1c > 53 mmol/mol). Additionally, this research emphasized that the most significant correlation was observed between glutamine and BCAA/aromatic amino acids [58]. It is worth mentioning that metformin, a commonly prescribed medication for managing T2D, can have a significant impact on the levels of different metabolites, such as those involved in the tricarboxylic acid (TCA) cycle, urea cycle, glucose metabolism, and lipid metabolism [23].

Another study discovered that individuals without any underlying health conditions who consumed a single dose of metformin experienced significant changes in metformin-related metabolites, including hydroxyl-methyl uracil, propionic acid, glycerol-phospholipids, and eicosanoids. Furthermore, metformin has the potential to yield various positive effects by influencing essential biochemical pathways like lipid signaling, energy balance, DNA damage repair mechanisms, and the composition of the gut microbiota [67].

In this line, Copenhagen Insulin and Metformin Therapy trial (*n* = 370 individuals) found that metformin therapy is associated with decreased amino acids, including valine, tyrosine, and carnitine serum levels associated with insulin resistance and mitochondrial dysfunction. Although, this study could not identify the metabolites which predict the HbA1c-lowering outcome of metformin [24]. Alanine levels most strongly increase in metformin monotherapy or dual therapy with sulfonylurea groups. Regardless, BCAAs (Val, Leu, and Ile) and the Fischer ratio (BCAA/aromatic amino acid ratio) were increased in those treated with metformin [23,25].

Following treatment with the gliclazide modified release, a commonly prescribed medication for T2D, significant improvements in blood glucose levels and insulin sensitivity were observed. These improvements were accompanied by changes in various metabolic pathways, including the tricarboxylic acid (TCA) cycle, ketone body metabolism, lipid oxidation, branched-chain amino acid breakdown, and gut flora metabolism. Furthermore, a panel of biomarkers consisting of HbA1c, 5,8,11,14,17-eicosapentaenoic acid, methyl 8,11,14-eicosatrienoate, and methyl hexadecanoate demonstrated accurate predictive ability in determining the suitability of gliclazide treatment. This finding holds significance for personalized medicine approaches in managing T2D patients undergoing sulfonylurea therapy [26].

Metabolomics signatures of other glucose-lowering medications, such as rosiglitazone and pioglitazone, have also been reported previously [29,30,31]. There is one study showed that after 16 weeks of treatment with rosiglitazone compared to placebo in patients with T2D and coronary heart disease, the treatment significantly increased circulating glutamine and decreased lactate concentrations. On another note, rosiglitazone treatment was able to reverse more abnormal levels of metabolites, such as valine, lysine, glucuronolactone, urate, and octadecanoate [64,66]. *Post hoc* analysis of a randomized clinical trial also found that insulin sensitizer therapy (pioglitazone plus metformin) could reduce nine out of thirty-three amino acids and their metabolites measured compared to placebo treatment [31].

Finally, pharmaco–metabolomics studies indicated that treatment with glucagon-like peptide-1 receptor agonists (GLP-1 RAs) leads to multiple metabolic changes, particularly that liraglutide treatment led to changes in the lipid metabolism involving sphingolipids, including ceramides that are suggestive of a lower risk of atherosclerosis and cardiovascular diseases (CVD) [27,28]. In addition, sphingolipids, such as ceramides, also represent one of the major lipid classes that could be considered among future biomarkers and target interventions in T2D prevention and therapy. Sphingolipids play an essential structural role in cell membranes by modulating multiple cell functions, such as apoptosis and cell inflammation [39].

## 4. Conclusions and Future Perspective

The broad application of metabolomics makes it a promising tool not only in the biomarker discovery field but also in disease etiological research, though much needs to be done from both the research perspectives and its clinical applications. The knowledge of metabolomics is relatively new, especially as a biomarker to monitor responses to medications. In recent years, complementary to genetic studies, metabolomics has been among the most popular and powerful tools for T2D diagnosis and prognosis. High throughput targeted and untargeted metabolomics approaches contributed to our fundamental understanding of the disease etiology and trajectories by promoting novel biomarkers. Metabolomics can be used to design new tests for diagnosis and treatments for T2D- to aid in identifying new drug targets and help understand the mechanisms behind T2D. Integrating metabolomics and clinical data in risk prediction models will add value; however, this requires further optimization and validation before they can be introduced into clinical practice. As the field advances, we anticipate that analysis will become more standardized.

Future studies combining metabolomics and other omics layers, such as genomics, transcriptomics, proteomics, and gut microbiota, will likely further elucidate the role of the identified metabolites in the pathogenesis of T2D and, more importantly, their potential in the diagnosis and management of diabetes. Additionally, metabolomics is an incredibly fast-growing field, and due to the growing availability of metabolomics data in prospective studies, comprehensive reviews, and analysis of a meta-analysis of serum, plasma, and urine, identifying metabolomics’ signature of T2D etiology and response to medications is necessary.

## Figures and Tables

**Table 1 genes-14-01464-t001:** An overview of metabolomics studies of type 2 diabetes.

Metabolite Type	Direction of Association with Type 2 Diabetes	References
Amino acids	BCAAs (Isoleucine, Leucine, Valine) (↑) AAAs (Phenylalanine, Tyrosine) (↑) Alanine (↑) Glutamate (↑) Methionine (↑) Histidine (↑) lysine (↑) Glycine (–) Glutamine (↓) 2-hydroxybutyrate (↑) 2-aminoadipate (↑)	[11,12,13]
Lipids	Lipoproteins HDL-C (↑) Triglyceride (↑) Glycerolipids Triacylglycerol (↑) Triacylglycerol (↑) Ceramides Dihydroceramide (↑) Phospholipids Phosphatidylcholine (↓) Di-acyl-phospholipids (↑) Lysoalkylphosphatidylcholine (↑) Lysophosphatidylcholine (↑) Alkyl-acyl phosphatidylcholines (↓) (lyso)phosphatidylethanolamines (↑)	[14,15] [16,17] [16,18,19,20,21]
Carbohydrates	Sugar monomer Mannose (↑) Treehouse (↑) Glucose (↑) Hexose (↑) Arabinose (↑) Fructose (↑) Glycolipid (↑) Polyol 1,5-anhydroglucitol(↓)	[16] [22]

The table summarizes previous studies which have investigated the association of metabolites with T2D. (↑), positive association (e.g., higher metabolite, higher risk); (–), controversial; (↓), inverse association (e.g., lower metabolite, lower risk) with prediabetes traits or type 2 diabetes.

**Table 2 genes-14-01464-t002:** An overview of metabolomics alteration by glucose-lowering medications.

Medication	Metabolites Alteration by Antidiabetic Therapy	References
Metformin	Tricarboxylic acid (TCA) cycle/Urea cycle/Hydroxyl-methyl uracil Glucose/Glycerol-phospholipids Propionic acid/Eicosanoids Valine/Tyrosine/Carnitine serum/BCAAs (Isoleucine Leucine Valine)	[23,24,25]
Gliclazide	Tricarboxylic acid (TCA) cycle/ketone body metabolites/methyl hexadecanoate lipid oxidation/5,8,11,14,17-eicosapentaenoic acid/methyl 8,11,14-eicosatrienoate BCAAs	[26]
Liraglutide	Sphingolipids (ceramides)	[27,28]
Rosiglitazone	Glutamine/Lactate/Valine/Lysine Glucuronolactone/urate/Octadecanoate	[29,30]
Pioglitazone	Clustered AA and metabolite pairs: (i) phenylalanine/tyrosine (ii) citrulline/arginine (iii) lysine/α-aminoadipic acid	[31]

## Data Availability

Not applicable.

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
