# Peer review of "Exploring Metabolomic Patterns in Type 2 Diabetes Mellitus and Response to Glucose-Lowering Medications—Review"

_genes, 2023, doi:10.3390/genes14071464_

Round 1
Reviewer 1 Report
Shahisavandi et al. present a review on the importance of metabolomics for diagnosis on type 2 diabetes. The review is well structured and gives a clear overview on advances in the approach of diabetes diagnosis and disease classification.
I have only some minor comments:
1. Please read through the manuscript, few sentences are not clear enough (e.g. the first sentence in the Introduction)
2. A figure showing for e.g. the main omics approaches in relation to the possible clinical benefit for the patient would be of clear benefit for the reader.
3. Please extend the discussion on the possible influence on patient outcome.
The manuscript needs correction by a native speaker.
Author Response
July 13th, 2023
Dear Editor,
Thank you for your time and effort in reviewing our manuscript. We appreciate the helpful comments and suggestions provided by the reviewers, which have significantly improved the quality of our work. We have carefully addressed the points that the reviewers raised and made the necessary revisions to the manuscript. Please find below a detailed response (in blue) to each of the points:
Sincerely,
Also, on behalf of all authors
Fariba Ahmadizar, PharmD, MSc, PhD
Reviewer #1:
I have only some minor comments:
- Please read through the manuscript, few sentences are not clear enough (e.g. the first sentence in the Introduction)
In the revised version of our manuscript, we have addressed this and made clarifications.
- A figure showing for e.g. the main omics approaches in relation to the possible clinical benefit for the patient would be of clear benefit for the reader.
Thanks for your suggestion; however, we have two tables showing metabolites and their associations with the outcomes of interest, which covers your suggestion.
- Please extend the discussion on the possible influence on patient outcome.
In the revised version of our manuscript, we have extended the discussion and addressed the possible influence on patient outcomes.

Reviewer 2 Report
This is a review article, which aims to discuss the role of metabolomics as potential novel biomarkers for the T2D diagnosis and response to glucose-lowering medications.
Generally, the topic is quite interesting, and the authors have in depth knowledge. They have used the appropriate manuscript design and presentation. The findings are well-presented, clear, and easy to understand. Overall, the manuscript is well written and structured. Thus, I think it would make a nice addition to Genes as a review article.
However, the following points should be generally considered, thus minor revision is demanded.
1. Line 31; Kindly consider to avoid characterizing patients as diabetic. Better to use the term patients with type 2 diabetes (T2D).
2. Line 51; Small discussion on the field of metabolomics assessment with these techniques would be a nice addition.
3. Line 57; The part of omics needs to be further discussed.
4. Line 79; Which metabolites specifically? Please report them through the manuscript.
5. Line 106-107; Kindly report the possible factors contributing to this finding.
6. Line 190; Kindly report the main metabolites with the strongest outcomes/evidence.
7. Line 227-230; Kindly provide more data about GLP-1RAs, as well as SGLT-2is and DPP-4is, if also studied.
Author Response
July 13th, 2023
Dear Editor,
Thank you for your time and effort in reviewing our manuscript. We appreciate the helpful comments and suggestions provided by the reviewers, which have significantly improved the quality of our work. We have carefully addressed the points that the reviewers raised and made the necessary revisions to the manuscript. Please find below a detailed response (in blue) to each of the points:
Sincerely,
Also, on behalf of all authors
Fariba Ahmadizar, PharmD, MSc, PhD
Reviewer #2:
- Line 31; Kindly consider to avoid characterizing patients as diabetic. Better to use the term patients with type 2 diabetes (T2D).
We thank the reviewer for this comment; we have applied this in the entire manuscript.
- Line 51; Small discussion on the field of metabolomics assessment with these techniques would be a nice addition.
We appreciate the reviewer for this comment.
In the revised version of our manuscript, we have added the following statement to page 4.
” Techniques related to metabolomics assessment usually include nuclear magnetic resonance (NMR), liquid chromatography-mass spectrometry (LC-MS), and gas chromatography-mass spectrometry (GC-MS)[8]. These techniques are used to identify and quantify metabolites from a biological sample. The recent rapid development of a range of analytical platforms, including GC, High performance liquid chromatography (HPLC), Ultra-performance liquid chromatography (UPLC), Capillary electrophoresis (CE) coupled to MS and NMR spectroscopy, could enable separation, detection, characterization and quantification of such metabolites and related metabolic pathways[9].”
- Line 57; The part of omics needs to be further discussed.
We are thankful for this comment. In the revised manuscript, we enhanced this part of the discussion as follows (page 10):
“Complex interactions between genetic and environmental factors influence the metabolite profile of an individual. GWAS have been used to explore the impact of genetic variation on plasma metabolites. The identified genetic variants associated with metabolite levels, particularly enzymes and carriers involved in processes like β-oxidation, fatty acid and phospholipid biosynthesis, and amino acid metabolism. Notably, these genetic loci explain a significant portion of the variance in metabolites, highlighting the role of genetics. Furthermore, specific genetic variants have been linked to glycine, serine, and betaine levels, although their connection to diabetes-related traits remains unclear[50].”
“Environmental factors such as diet, activity, medication, and the microbiome contribute to the complexity of the metabolome. Diet, physical activity, gender, and age influence metabolomic profiles and should be considered in study design and interpretation. The gut microbiome plays a significant role in host metabolism and metabolomics profiles, with obesity and T2D associated with altered microbial profiles and reduced diversity. Microbial populations can impact host metabolism, intestinal development, and insulin secretion and contribute to metabolic disorders. Microbiota transplantation studies in mice and limited human data suggest the potential for improving metabolic health through modulation of the microbiome [51-53].”
- Line 79; Which metabolites specifically? Please report them through the manuscript.
In the revised manuscript, we have clarified this section by adding detailed metabolites (pages 5 and 6).
“A recent systematic review and meta-analysis investigated the relationship between 412 metabolites present in blood or urine metabolites and incident T2D. The study encompassed a comprehensive analysis of 61 individual studies (71,196 participants and 11,771 incident T2D). By aggregating the data from these studies, the researchers aimed to gain a deeper understanding of the association between specific metabolites and the risk of developing T2D. This study indicates several blood metabolites, including lipids and carbohydrates, are associated with T2D development [13]. Besides that, the increased levels of certain amino acids were associated with insulin resistance in non-obese individuals, suggesting that amino acid imbalances play a role in this condition, independent of commonly accepted risk factors like circulating fatty acids and inflammatory cytokines [17].”
“Highlighting the significance of understanding the metabolic disturbances in T2D and its complications, recent advancements in metabolomics have enabled the identification of circulating biomarkers associated with T2D even before its onset. These biomarkers offer potential applications in screening, diagnosis, and prognosis while also providing insights into the underlying pathways involved in T2D development. Integrating the omics approaches with genomics can uncover causal associations, although careful utilization of these methods is necessary due to their limitations [12].”
- Line 106-107; Kindly report the possible factors contributing to this finding.
Accordingly, we further discussed these factors as increased levels of certain amino acids were associated with insulin resistance in non-obese individuals, suggesting that amino acid imbalances play a role in this condition, independent of commonly accepted risk factors like circulating fatty acids and inflammatory cytokines [17].
- Line 190; Kindly report the main metabolites with the strongest outcomes/evidence.
We thank the reviewer for this comment. According to an updated meta-analysis of the case-cohort studies (71,196 participants and 11,771 T2D cases/events), higher carbohydrate metabolites, including mannose and trehalose, are also associated with an increased risk of T2D. Besides that, higher levels of glycine, glutamine, betaine, indole propionate, and(lyso)phosphatidylcholines were associated with lower T2D risk (hazard ratio 0.69–0.90) [6].
- Line 227-230; Kindly provide more data about GLP-1RAs, as well as SGLT-2is and DPP-4is, if also studied.
This is indeed very interesting and would add to the information provided by our review. However, our search did not find any relevant study investigating metabolites associated with response to these drugs.
